# Improving qubit coherence using closed-loop feedback

Antti Vepsäläinen [1✉], Roni Winik [1], Amir H. Karamlou [1,2], Jochen Braumüller [1], Agustin Di Paolo [1], Youngkyu Sung [2], Bharath Kannan [2], Morten Kjaergaard[1,3], David K. Kim[4], Alexander J. Melville[4], Bethany M. Niedzielski [4], Jonilyn L. Yoder[4], Simon Gustavsson [1] & William D. Oliver[2,4]

Superconducting qubits are a promising platform for building a larger-scale quantum processor capable of solving otherwise intractable problems. In order for the processor to reach practical viability, the gate errors need to be further suppressed and remain stable for extended periods of time. With recent advances in qubit control, both single- and two-qubit gate fidelities are now in many cases limited by the coherence times of the qubits. Here we experimentally employ closed-loop feedback to stabilize the frequency fluctuations of a superconducting transmon qubit, thereby increasing its coherence time by 26% and reducing the single-qubit error rate from $(8.5 \pm 2.1) \times 10^{-4}$ to $(5.9 \pm 0.7) \times 10^{-4}$. Importantly, the resulting high-fidelity operation remains effective even away from the qubit flux-noise insensitive point, significantly increasing the frequency bandwidth over which the qubit can be operated with high fidelity. This approach is helpful in large qubit grids, where frequency crowding and parasitic interactions between the qubits limit their performance.

[1] Research Laboratory of Electronics, Massachusetts Institute of Technology, Cambridge, MA, USA. [2] Department of Electrical Engineering and Computer Science, Massachusetts Institute of Technology, Cambridge, MA, USA. [3] Center for Quantum Devices, University of Copenhagen, Copenhagen, Denmark. [4] MIT Lincoln Laboratory, Lexington, MA, USA. ✉email: apvepsala@gmail.com

Igh-fidelity single- and two-qubit gates are a prerequisite for high-depth circuits and quantum error correction. For many qubit modalities, including superconducting qubits, the qubit frequencies and their controls are subject to temporally correlated noise—most notably $1/f$-type noise[1]—resulting in correlated errors and slow drifts in frequency.

Closed-loop feedback control is ubiquitously used in a wide variety of engineering applications. There is an increasing number of experiments where feedback is used to modify the evolution of quantum systems, such as stabilizing the motion of the atoms in optical cavities[2], cooling quantum mechanical resonators[3], or stabilizing Rabi oscillations in superconducting qubits[4]. These experiments are based on continuous monitoring of the system, which inevitably decoheres to the quantum state. In ref. [5], a method based on interleaving the probing sequences with separate periods of time used for the computation was introduced and demonstrated to mitigate the effect of slow magnetic field fluctuations. In ref. [6], it was further shown that this method can be used to decouple an electron spin from low-frequency magnetic field fluctuations, resulting in improved coherence times. Ref. [7] employs a slightly different approach in a trapped-ion system, where spectator qubits are used to probe spatially correlated errors in the control laser amplitude and targeting.

The dominant source of decoherence in superconducting qubits is typically either charge noise or flux noise but depending on the qubit design also photon shot noise[8] or Bogoliubov quasiparticles may contribute[9]. These noise sources are primarily intrinsic and local to the device[10,11]—as opposed to noise in the control electronics—though recently there has been some evidence of correlated noise between the qubits[12–14]. In our experiment, we employ flux-tunable transmon qubits[15], which are widely used in contemporary superconducting quantum processors[16,17]. Due to their design, transmons are mostly insensitive to charge noise[15], but suffer from noise in magnetic flux, which is used to tune their frequency. In order to protect the qubits from flux noise, these qubits are typically operated at bias points where their frequency is first-order insensitive to small changes in flux, colloquially referred to as a sweet spot, see Fig. 1a. However, in many architectures, the qubits cannot be operated at the sweet spot indefinitely while performing logic operations[18,19]. Additionally, it is necessary to operate some of the qubits away from their sweet spots, in part due to parasitic couplings to two-level fluctuators with frequencies near the sweet spot[20–23] or couplings to other qubits or their higher excited states in the quantum processor[17].

Here, we use active feedback control to stabilize the frequency drift of a flux-tunable superconducting transmon qubit and thereby suppress its temporal frequency fluctuations. This results in improved coherence times and gate fidelities, which remain stable over extended periods of time. We demonstrate that using the feedback protocol to suppress low-frequency noise enables gate fidelities exceeding 99.9% even far away from the flux sweet spot, greatly extending the range of available operation frequencies for such qubits.

## Results

The feedback protocol we employ consists of three phases that are continuously repeated, see Fig. 1b. In the probing phase, the qubit frequency is estimated using a simple single-qubit frequency estimation algorithm. After the frequency of the qubit is estimated, the magnetic flux through the qubit SQUID loop is adjusted to set the qubit frequency to its target value. This is followed by a computation phase, where an algorithm can be run with a freshly stabilized qubit.

The probing phase consists of $N$ repeated Ramsey measurements. For each of those measurements, the qubit is first prepared in a superposition state $|\psi\rangle = (|0\rangle + |1\rangle)/\sqrt{2}$ using a $\pi/2$ rotation around the $y$ axis of the Bloch sphere. The state preparation is followed by a period of free evolution for a duration $\tau$, during which the qubit state acquires a phase $\phi = 2\pi \int_0^\tau \delta_q(t)dt$, where $\delta_q(t) = f_d - f_q(t)$ is the detuning between a microwave drive frequency $f_d$ defining a rotating reference frame, and $f_q(t)$ is the fluctuating qubit frequency in the presence of noise. A second $\pi/2$ pulse is then applied around the $x$ axis, and the state of the qubit is measured using dispersive readout[24]. To simplify the feedback protocol, we make a quasi-static approximation and assume that the qubit frequency remains constant within one frequency estimation experiment—$N$ Ramsey measurements—but may fluctuate between the experiments[25]. With this assumption, the probability of measuring the qubit in the excited state is given by

$$p_1 = \frac{1}{2} + \frac{1}{2}\cos(2\pi\delta_q\tau - \pi/2), \tag{1}$$

which can be inverted to yield the frequency shift

$$\delta_q = \frac{\pm \arccos(2p_1 - 1) + 2\pi k + \pi/2}{2\pi\tau}, \tag{2}$$

where $k$ is an integer. Equation (2) is a one-to-one mapping from $p_1$ to the frequency detuning $\delta_q$ over the domain $\delta_q \in [-\frac{1}{4\tau}, \frac{1}{4\tau}]$. This implies the fluctuations in $\delta_q$ need to be within $\pm\frac{1}{4\tau}$ between the estimation steps, ~70 μs in our experiment.

The qubit excited-state probability estimator $\hat{p}_1 = \frac{1}{N}\sum_{i=0}^N q_i$ is calculated from the measurement record $q_i$ of $N = 20$ repetitions of the Ramsey sequence, providing an estimate for the frequency detuning of the qubit, $\hat{\delta}_q$, by substituting $\hat{p}_1$ into Eq. (2). If in the previous measurement the qubit was measured to be in the excited state, we virtually reset the qubit state for the current repetition by flipping $q_i$[26,27]. The duration of a Ramsey measurement is $T = 3.5$ μs, comprising the phase accumulation time $\tau = 1.25$ μs, the readout duration of 750 ns, and the combined resonator reset time and overhead from the electronics, 1.5 μs. Thus, a single round of frequency estimation takes $T_N = NT = 70$ μs in our implementation. The frequency estimation is not able to detect fluctuations that occur faster than the repetition period of the feedback, ultimately limiting the effective bandwidth of the noise suppression. Next, we investigate how the feedback reduces the noise spectral density of the qubit.

**Noise spectral density**. To estimate the noise power spectral density affecting the qubit frequency, we first bias the qubit away from the sweet spot by 0.11 flux quanta $\Phi_0 = h/2e$, at a transition frequency $f_q = 4.69$ GHz that is sensitive to flux noise (Fig. 1a), and repeatedly estimate its frequency $10^5$ times. The blue dots in Fig. 1d show the power spectral density of the measured qubit frequency fluctuations subject to flux noise. The measured power spectral density follows a power law (green solid line) $S_{f_q f_q}(f) = A_{f_q}\left(\frac{1\,\text{Hz}}{f}\right)^\alpha \approx 27.3 \times 10^6\,\text{Hz}^2/\text{Hz} \times \left(\frac{1\,\text{Hz}}{f}\right)^{0.8}$, but starts to deviate at frequencies above 500 Hz for $N = 20$ and $\tau = 1.25$ μs. This is due to noise added by the finite number of samples in the estimate $\hat{\delta}_q$ and can be approximated as

$$\delta\hat{\delta}_q \approx \frac{1}{2\pi\tau\sqrt{N}}, \tag{3}$$

see Supplementary material for the derivation. The statistical sampling noise is modeled as Gaussian white noise with

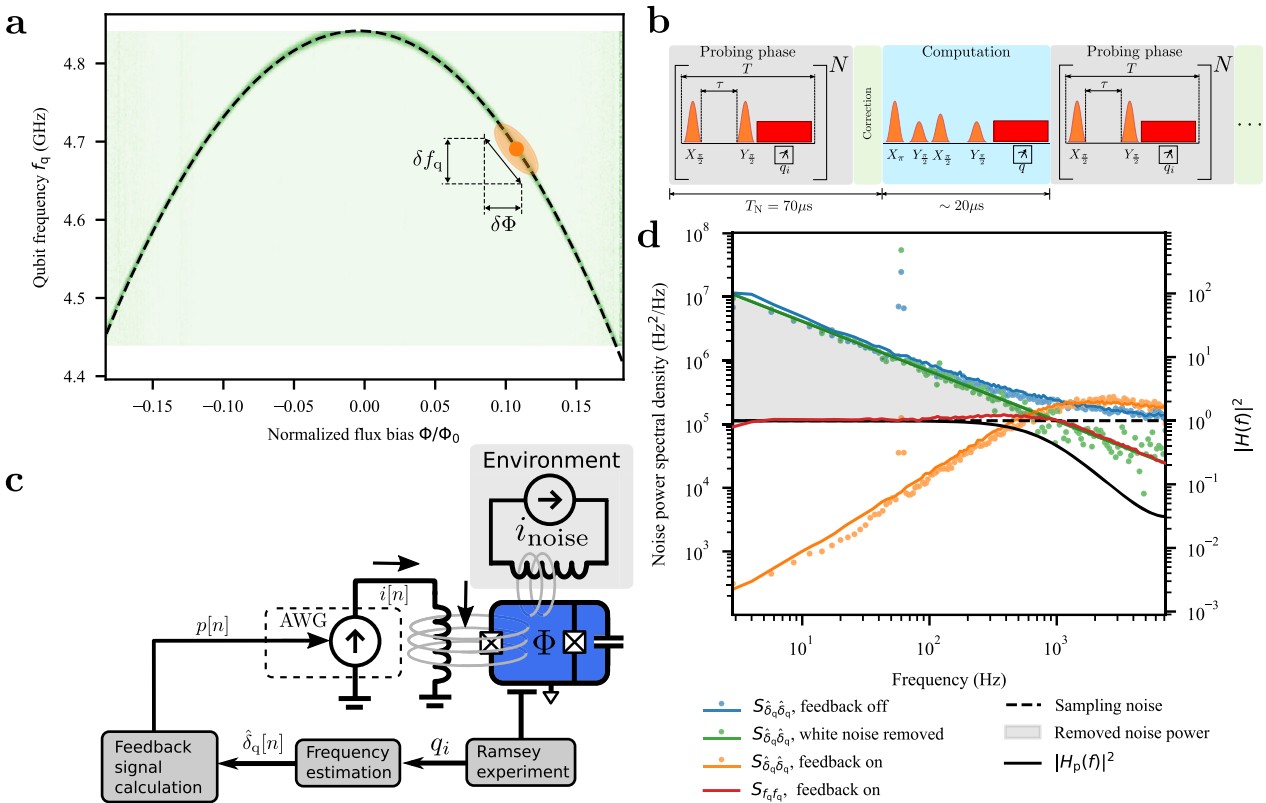

**Fig. 1 Qubit frequency power spectral density. a** The spectrum of the qubit as a function of flux bias. The orange dot shows the flux bias point at which the qubit is operated. **b** Schematic of the frequency estimation pulse sequence (gray) interleaved with the sequence used for computation (light blue). The qubit frequency is adjusted in between (green). **c** Schematic of the feedback loop. The measurement record $q_i$ of a repeated Ramsey experiment is used to estimate the qubit frequency offset $\hat{\delta}_q[n]$ subject to a noisy environment $i_{noise}$. The feedback signal $p[n]$ controls an AWG that produces current $i[n]$ to cancel the fluctuations in the qubit frequency. **d** Power spectral density of the qubit frequency fluctuations. The blue dots (line) show the measured (simulated) spectral density of the qubit frequency fluctuations estimated from $N = 20$ Ramsey experiments $S_{\hat{\delta}_q\hat{\delta}_q}(f)$, limited by the statistical sampling noise (dashed black line). The spectrum with the sampling noise suppressed using cross-correlation between consecutive samples is shown with green dots along with a fit (green line). Orange dots (line) are the measured (simulated) power spectral density of the error signal $\hat{\delta}_q[n]$ with the feedback activated. The simulated power spectral density of the actual qubit frequency fluctuations $f_q(t)$ is shown with a solid red line. The gray background describes the noise power removed by the feedback. The frequency response of the feedback signal is overlaid in the plot with a solid black line.

an upper cutoff given by the duration of the frequency estimation[28],

$$
S_{est}(f) = \begin{cases} \frac{T}{2\pi^2\tau^2}, & 0 \le f \le \frac{1}{2NT}, \\ 0, & \text{otherwise}, \end{cases} \quad (4)
$$

shown with a dashed black line in Fig. 1d. The sampling noise can be suppressed in post-processing by cross-correlating time-shifted measurement traces as demonstrated in ref. [28]. The results of applying this protocol are shown with green dots in Fig. 1d. With the sampling noise suppressed, the measured power spectrum fits well to the power-law across the whole bandwidth. There is an additional peak at 60 Hz corresponding to the noise from the main power. The sampling noise suppression is not used in the real-time feedback signal calculation due to the small amount of samples available at the time of computation.

Next, we turn on the feedback to reduce the fluctuations in the qubit frequency. We aim to minimize the deviation of the qubit frequency $f_q(t)$ from the target frequency $f_d$ by using the offset $\hat{\delta}_q[n]$ as the error signal in the feedback loop, see Fig. 1c for the schematic of the signal flow. Here we use $n$ to number each time feedback is applied, sampled at times $t_n = nT_N$. In practice, the sampled error signal represents the average of the qubit frequency fluctuation during the sampling period $T_N$, limiting the maximum bandwidth of the feedback to $1/(2T_N) \approx 7$ kHz if the

time spent on the interleaved computation step is omitted. The error signal $\hat{\delta}_q[n]$ is multiplied by a controllable gain $G$ and fed into an accumulator that controls the feedback signal, $p[n] = p[n-1] + G\delta_q[n]$. We deliberately set $G = 0.35$ to reduce the bandwidth of the feedback frequency response $H_p(f)$, shown with solid black line in Fig. 1d, to be less affected by the statistical sampling noise, see Supplementary material for additional details.

The output of the accumulator $p[n]$ is scaled and converted to an arbitrary waveform generator voltage that drives the current responsible for creating a magnetic flux through the qubit loop, adjusting its transition frequency. The feedback significantly reduces the noise spectral density of the error signal $\hat{\delta}_q[n]$, shown with orange dots in Fig. 1d.

After closing the feedback loop, the error signal $\hat{\delta}_q[n]$ provides only indirect information about the actual qubit frequency fluctuations, which are also affected by the frequency response of the feedback (see Supplementary material for theoretical analysis). Thereby, we employ a simulation to assess the impact of the feedback transfer function on stabilizing the qubit frequency $f_q(t)$. We use the fitted noise spectral density as the starting point of the simulation (green solid line in Fig. 1d) to generate time traces of the fluctuating qubit frequencies. Using the same parameters as in the experiment, we simulate the estimation of the qubit frequency and the feedback, which results in a time

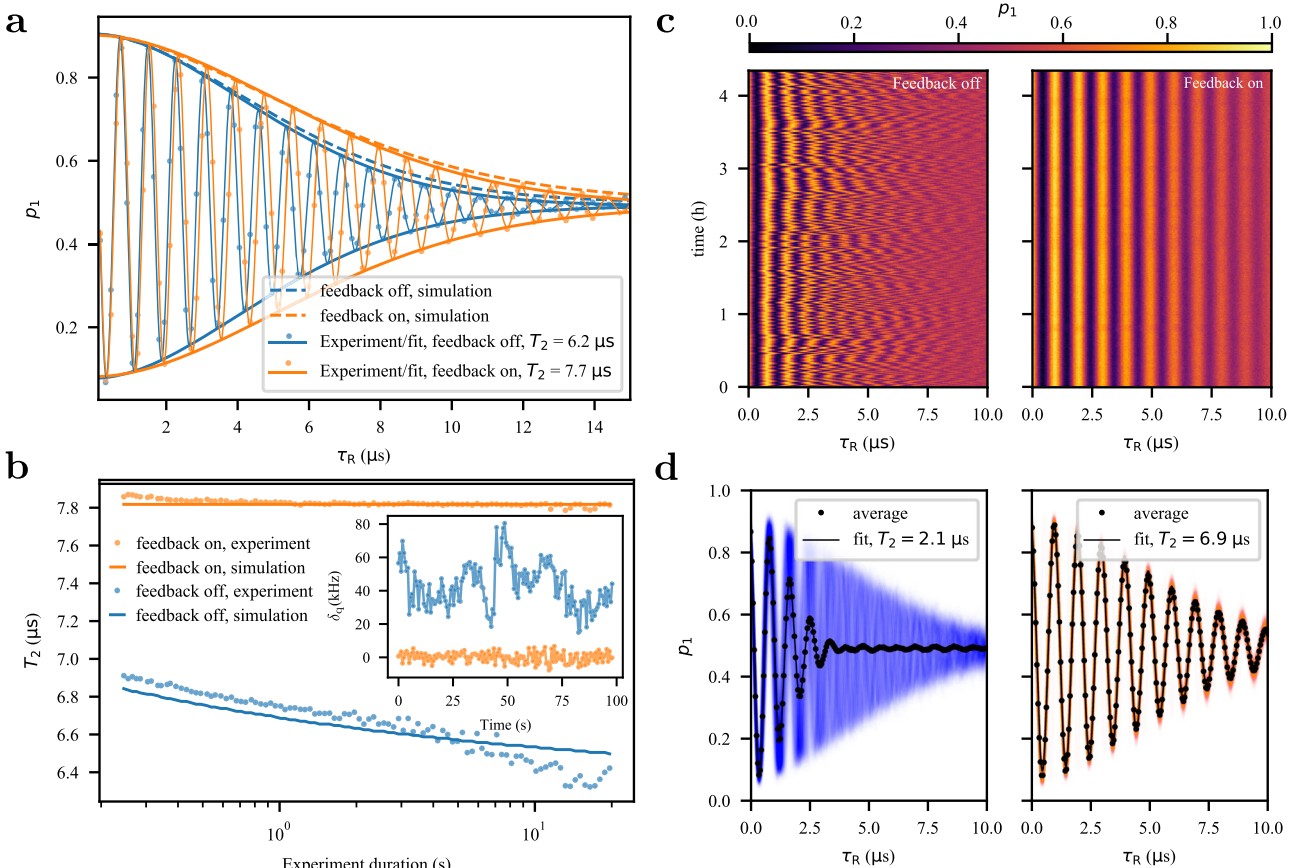

**Fig. 2 Improvement in qubit coherence and stability. a** The coherence time $T_2$ of the qubit is measured by interleaving a Ramsey measurement with feedback sequences (orange dots), compared to when the feedback is off (blue dots). The solid lines are a fit for the data. Dashed lines show a simulation of the decay envelope assuming the noise spectral densities shown in Fig. 1d. **b** The measured coherence time of the qubit as a function of the duration of the Ramsey experiment is shown with the feedback (orange dots) and without (blue dots). The solid lines are the expected coherence times based on the measured noise spectral density. The inset shows the measured deviation of the qubit frequency from the target frequency during the experiment. Each point is calculated using 50 averages of the Ramsey trace. **c** shows Ramsey experiment repeated for 4 hours with (right) and without (left) feedback. With the feedback, the qubit frequency is stable during the whole duration. **d** Averaging the data in (**c**) results in a significant reduction in qubit coherence if feedback is not used (left). Using the feedback, the inferred coherence is not affected by the measurement duration (right). The coherence time of a single Ramsey trace here is slightly lower than in (**a**) due to the smaller value of $\tau = 500$ ns used in the feedback.

trace of estimated qubit frequencies. The power spectral density of the simulated qubit frequency estimation without the feedback is shown with a solid blue line in Fig. 1d, matching the experiment almost perfectly. The solid orange line shows the simulated qubit frequency estimates when the feedback is turned on, again matching well with the experiment. Finally, using the simulation we can calculate the power spectral density of the real qubit frequency fluctuations when the feedback is applied, shown with a solid red line in Fig. 1d. The total noise power mitigated by the feedback protocol is indicated by the shaded area in Fig. 1d.

**Qubit coherence.** Next, we interleave the frequency estimation sequences with a Ramsey experiment to demonstrate that the lower noise power increases the qubit coherence time $T_2$, see schematic in Fig. 1b. We apply two $\pi/2$ pulses around the $y$-axis of the Bloch sphere, and sweep the delay between the pulses, $\tau_R$. Figure 2a shows the qubit excited-state population $p_1$ going through the Ramsey oscillations, first without the feedback, and then with the feedback turned on. For a Gaussian-distributed noise process, the envelope of the Ramsey oscillations decays as[29]

$$\chi_R(t) = \exp\left[-2t^2\pi^2 \int_{f_0}^{\infty} S_{f_q f_q}(f)\,\text{sinc}^2(\pi f t)\,\mathrm{d}f\right], \quad (5)$$

where $S_{f_q f_q}(f)$ is the unilateral power spectral density of the frequency fluctuations and $f_0$ is the lower cutoff frequency equal to the inverse of the total duration of the experiment. We fit the experimental data to an oscillating function with a decay envelope corresponding to a power spectral density of $1/f$ noise and extract the coherence time from when the decay envelope drops below $1/e$ from its original value at $t_R = 0$. The feedback increases the coherence time from $T_2 = 6.2\,\mu s$ to $T_2 = 7.7\,\mu s$, or 26%. The observed improvement in coherence time was consistent across several repetitions of the experiment. The fitted decay envelope can be compared to a theoretical estimate obtained by directly substituting the measured power spectra to Eq. (5), shown with dashed lines in Fig. 2a, closely matching experimentally observed decay envelopes. The only fit parameter is the qubit state initialization fidelity, 92%, which scales the decay envelope amplitude. This confirms that the decoherence of the qubit is well described by the measured power spectral density of the frequency fluctuations.

The inferred coherence time of the qubit depends on the total duration of the Ramsey experiment through the cutoff frequency $f_0$ in Eq. (5). In Fig. 2b, the coherence time $T_2$ is evaluated using different numbers of averages to calculate $p_1$, thereby changing the total duration of the single Ramsey experiment and the cutoff

frequency $f_0$. The data from the experiment is collected only once, and is sectioned to different numbers of averages as post-processing. Without the feedback, the coherence time gradually decreases as the duration of the experiment increases. This is due to the increased noise power at lower frequencies for $1/f$ noise. When the feedback is activated, the overall coherence time is increased—similar to the experiment shown in Fig. 2a—and remains constant independent of the cutoff frequency $f_0$ due to the elimination of the low-frequency noise. Moreover, the qubit frequency remains stable during the measurement, as inferred from the frequency of the Ramsey oscillations during the experiment, shown in the inset of Fig. 2b. The qubit frequencies in the inset are evaluated from a running average of 50 Ramsey traces.

The stability of the qubit frequency can be maintained for hours using the feedback protocol as demonstrated in Fig. 2c. There, Ramsey experiments are repeated for more than four hours, either with the feedback turned off (left panel) or on (right panel). When the measured qubit excited-state populations are overlaid (Fig. 2d), the Ramsey oscillations without the feedback are blurred due to the constant fluctuation in the qubit frequency, whereas with the feedback the oscillations are clearly visible. The coherence time extracted from the average of all uncorrected (feedback off) Ramsey experiments is only $T_2 = 2.1$ μs, compared to $T_2 = 6.9$ μs with the feedback.

Thus far, we have operated the qubit at a fixed flux bias point. We next demonstrate that the feedback protocol improves the coherence time for a range of flux biases. The sensitivity of the qubit to the flux noise is determined by the curvature of its frequency spectrum with respect to flux bias, allowing us to probe the efficiency of the feedback at various noise levels and intrinsic coherence times. We evaluate the pure dephasing rate of the qubit at 11 different bias points, first without the feedback and then with it on. The pure dephasing rates are extracted from the decay of the Ramsey oscillations by subtracting the effect of the energy-relaxation rate[10]. The measured dephasing rates (Fig. 3) are

lowest close to the flux sweet spot and gradually increase away from this spot as the qubit sensitivity to the noise increases. In the limit where decoherence is dominated by flux noise, Eq. (5) can be used to show that there is an (almost) linear dependence between the dephasing rate $\Gamma_\phi$ and the flux sensitivity of the qubit[10,25,29],

$$\Gamma_\phi = k \left| \frac{\partial f_q}{\partial \Phi} \right|. \tag{6}$$

We find the coefficient $k$ from a linear fit to the data in Fig. 3 and use the result to assess the impact of the feedback on mitigating the flux noise. Without the feedback $k = (58 \pm 1.1)\mu\Phi_0$, and reduces to $k = (48 \pm 1.4)\mu\Phi_0$ when the feedback is used, an improvement of 17%. This implies that the feedback effectively reduces the flux noise amplitude seen by the qubit.

The ultimate goal would be to suppress the low-frequency noise to the level that the coherence time measured using a spin-echo experiment—which is insensitive to low-frequency noise—would be equal to or higher than the coherence time in a feedback-stabilized Ramsey experiment. In Fig. 3, the dephasing rate measured from an echo experiment are shown with black dots, and the linear fit yields $k_E = (17 \pm 0.5)\mu\Phi_0$, which corresponds to the $1/f$-flux-noise amplitude of $\sqrt{A_\Phi} = (3.3 \pm 0.1)\mu\Phi_0$, see Supplementary material for additional information. The main reason why the impact of the noise on the echo experiment is lower than the on feedback-stabilized Ramsey experiment is the limited bandwidth of our feedback implementation. While the feedback efficiently suppresses the noise up to 1 kHz, the echo experiment is mostly insensitive to the noise below the inverse duration of a single echo experiment, here ~100 kHz. This suggests that by improving the feedback implementation, further improvements in the coherence time are attainable, see Supplementary material for analysis.

Unlike the spin-echo experiment, which is specifically designed to be insensitive to the low-frequency qubit frequency fluctuations, many quantum algorithms or sequences of quantum gates are highly sensitive to small deviations in the qubit frequency. While there exist several open-loop control strategies for minimizing gate sensitivity to noise at different frequencies[30,31], there is always an added cost in the duration of the gate sequence or complexity in calibration. The advantage of the feedback-based stabilization method is that no changes to the gate sequences or controls are required.

**Randomized benchmarking**. Next, we demonstrate that the feedback protocol improves the single-qubit gate fidelity in our device. We bias the qubit 400 MHz away from the flux sweet spot so that it is highly sensitive to flux noise and perform single-qubit randomized benchmarking[32] while stabilizing the qubit frequency with the feedback. Figure 4a shows the qubit excited-state population $p_1$ as a function of the number of Clifford gates in a random sequence, followed by a Clifford gate that would ideally bring the qubit back to the ground state. With the feedback off (blue) and on (orange), the experiment is repeated for 50 different random sequences, which are averaged and used to find the average error per gate[33]. The feedback reduces the average error per gate from $(8.5 \pm 2.1) \times 10^{-4}$ to $(5.9 \pm 0.7) \times 10^{-4}$, approaching the limit imposed by the energy-relaxation rate of the qubit, see Supplementary material for additional information. While such fidelities are commonly observed at or near the flux-insensitive point in many devices[33,34], it is unusual to see them so far from the sweet spot.

We attribute the reduction in the average gate error mostly to the improved stability of the qubit frequency. Without feedback, several of the random sequences show oscillating decay functions with the number of Clifford gates, indicative of coherent control

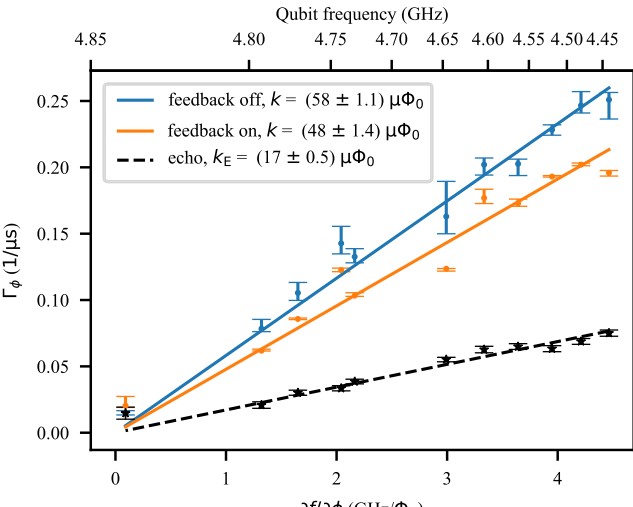

**Fig. 3 Qubit coherence at different bias points.** The qubit dephasing rate $\Gamma_\phi$ is evaluated at several different bias fluxes. Qubit's sensitivity to flux noise increases further away from the sweet spot, resulting in reduced dephasing times (increased dephasing rates). The error bars show 68% confidence intervals for the median of the dephasing rates measured 60 times. The solid lines show a linear fit to the dephasing rates with respect to the curvature of the qubit spectrum with respect to flux. The black stars show the dephasing rates extracted from a spin-echo experiment, used as a reference.

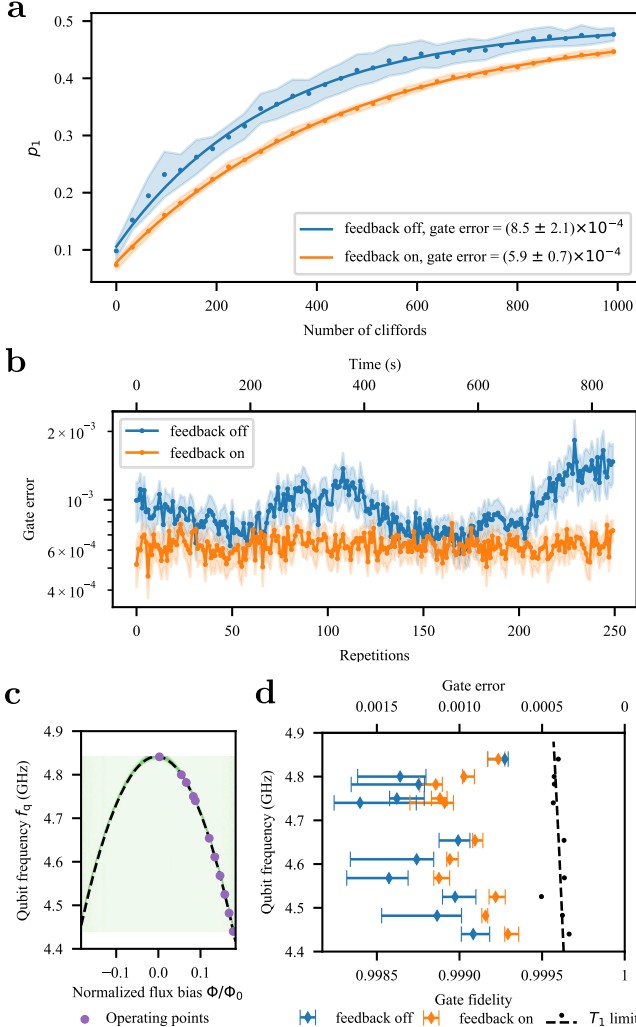

**Fig. 4 Randomized benchmarking. a** Randomized benchmarking of the single-qubit gates at $f_q = 4.44$ GHz. The orange (blue) dots show the average of 50 realizations of the random Clifford sequences with (without) the feedback. The shaded area shows the 68% confidence interval for the average trajectory. The gate error is extracted from the fit to the data (solid line). **b** Randomized benchmarking is repeated 250 times with the feedback (orange dots) and then without (blue dots). The shaded area shows the 68% confidence interval for the fitted gate error. **c** The qubit spectrum and the operating points at which the gate errors are evaluated in (**d**). **d** Randomized benchmarking at several different qubit bias fluxes. Black dots show the estimated coherence limit for the gate fidelities, inferred from the energy-relaxation time $T_1$ of the qubit[33]. The dashed black line shows a linear fit to the estimated coherence limits at the different operating points. The error bars are the 68% confidence intervals for the median gate error.

errors, such as the quasi-static shifts in the qubit frequency due to the low-frequency noise. In contrast, with the feedback activated, the function monotonically decays for all random sequences, something that is typical for incoherent errors[35–39]. This observation is supported by the experiment in Fig. 4b where we show the gate errors inferred from randomized benchmarking experiments repeated 250 times with seven different realizations of a random gate sequence, for a total duration of 840 s. Without the feedback, we observe a significant drift in the gate errors. With feedback turned on, the gate errors remain consistently low, indicating that the feedback successfully stabilized the drift and fluctuations in the qubit frequency. This is also manifested in lowered uncertainty in the fitted gate error in Fig. 4a.

We observe a similar improvement in gate fidelity for all flux bias points with the exception of the sweet spot, see Fig. 4c, d. At each flux point, we repeated the randomized benchmarking experiment 10 times with and without feedback, re-calibrating the qubit frequency between every repetition of the randomized benchmarking experiment. In contrast to the coherence time $T_2$— which is the highest at the sweet spot and then consistently decreases as the flux noise sensitivity increases away from the sweet spot—the highest gate fidelities are in fact measured at the most flux-sensitive point we investigated. We attribute the increasing trend in the fidelities at lower frequencies to the higher energy-relaxation time $T_1$ of the qubit. In addition, spectrally moving parasitic two-level fluctuators reduced the best achievable single-qubit gate fidelities at certain bias points. Owing to the coupling to the qubit, the changes in the frequencies of the two-level fluctuators caused shifts in the qubit frequency. As a result, at some bias points, there were wide variations in the measured gate fidelities between different experiments, which are responsible for the large error bars in Fig. 4c when the feedback was not used. The variation is significantly reduced by the feedback, as indicated by the repetition experiment shown in Fig. 4b and further verified in Fig. 4c. This highlights that the implemented feedback protocol can mitigate the impact of many sources of low-frequency noise, such as frequency shifts caused by two-level fluctuators, see Supplementary material for additional analysis.

## Discussion

In this work, we have implemented a closed-loop feedback protocol to stabilize the drift and fluctuations in the frequency of a superconducting transmon qubit. In the probing phase, we use repeated Ramsey experiments to estimate the qubit frequency and adjust the qubit frequency to cancel the measured frequency offset. The probing phase can be interleaved with a computational workload such as algorithm execution.

We have demonstrated that the feedback stabilizes the qubit frequency fluctuations even when the qubit is not operated at the noise-insensitive operation point. This leads to a reduction in the noise power observed by the qubit, resulting in improved coherence times and improved gate fidelities. The ability to operate qubits away from the protected bias point will help address the frequency crowding problem in large quantum processors by increasing the operable frequency band for the qubits. In addition, the increased operable frequency band helps to avoid spurious modes arising for example from two-level fluctuators.

Although in this work we have mostly focused on mitigating $1/f$-flux noise, the feedback algorithm is agnostic to the source of the noise in the qubit frequency. The proposed technique should work equally well on other low-frequency noise sources, such as charge noise or TLS induced shifts in the qubit frequency. Moreover, the feedback relies only on single-qubit operations which implies that it can be extended to multi-qubit systems without additional cost in time or complexity. As quantum processors grow in size, the proposed feedback algorithm could increase their reliability by continuously calibrating for drifts in the individual qubits.

## Data availability

The data that support the findings of this study may be made available from the corresponding authors upon reasonable request.

## Code availability

The code used for the analyses may be made available from the corresponding authors upon reasonable request.

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

## Acknowledgements

This work was supported in part by the U.S. Army Research Office (ARO) Grant W911NF-18-1-0411; by the ARO Multi-University Research Initiative W911NF-18-1-0218; and by the Assistant Secretary of Defense for Research and Engineering via MIT Lincoln Laboratory under Air Force Contract no. FA8721-05-C-0002. A.K. acknowledges support from the NSF Graduate Research Fellowship program. The authors thank Andy Ding for their comments on the manuscript.

## Author contributions

A.V., R.W., S.G., and W.D.O. conceived the project. A.V. performed the experiments. R.W. programmed the FPGA used for feedback. B.K. designed the sample. A.V. and R.W. analyzed the data with feedback from A.D.P., S.G., and W.D.O. J.B., A.H.K., Y.S., and M.K. contributed to the software infrastructure. D.K.K., A.J.M., B.M.N., and J.L.Y. fabricated the sample.

## Competing interests

A patent application for the feedback implementation is being prepared by A.V., R.W., S.G., and W.D.O. Other authors declare no competing interests.
