## [Peer Review File · Nature Communications]

REVIEWER COMMENTS

Reviewer #1 (Remarks to the Author):

In the present manuscript Authors report the experimental realization of closed-loop feedback to stabilize low frequency fluctuations of the frequency of a superconducting flux tunable transmon qubit. This is a crucial issue since limitation in coherence times of present day superconducting qubits is a key limiting effect towards the realization of larger scale quantum processors. One of the main issues are charge, flux and Josephson energy noise with $1/f$ spectrum, a problem investigated in a number of articles (see below for relevant references on this topic). The key results of the manuscript are the demonstration of 26% improvement of the decoherence time away from the flux sweet point and the 99.9% fidelity of single-qubit randomized benchmarking.

Original results are derived using appropriate experimental methods and data are analyzed applying established frequency estimation methods to the closed-loop feedback protocol consisting of three phases that are continuously repeated. Obtained results, potentially allowing for increasing the operable frequency band of qubits, may have relevant implications for addressing the frequency crowding problem in large scale quantum processors. The manuscript is certainly suitable for publication in Nat. Comm., but it may still be improved considering the following issues.

The closed-feedback approach is essentially a method to recalibrate the inhomogeneous broadening effect of the qubit splitting due to repeated runs of an experiment taking place in the presence of low-frequency noise. The quasi-static approximation captures this effect and results in short-times decoherence of Ramsey fringes with a characteristic non-exponential decay, depending on the operating point, and obtained originally in G. Falci et al Phys Rev Lett 94, 167002 (2005) and shortly later in Ref.[27] of the manuscript. In G. Falci et al, the improvement by recalibration was demonstrated and a systematic inclusion of the effect of higher-frequency noise components, active with feedback-on, was performed, leading to analytical predictions. I expect that the present work may be strengthened by the following analysis of experimental data.

- Case feedback off: the accuracy level of the present experiment may provide a benchmark of the power law to exponential quadratic decay predicted in G. Falci et. al, at- and away-from the flux optimal point. This analysis may also point out whether noise in the Josephson energy plays a role, as pointed out in Ref. [14] of the manuscript and which I would have expected to observe.

- Case feedback on: One of the key results of the present manuscript is the improvement of decoherence times away from the qubit flux-noise insensitive point obtained by closed-feedback.

This effect is analysed based on eq. (5) which holds under pure dephasing and it is used to fit data in Fig. 3 with a reasonable but not excellent agreement. Under the considered conditions the residual decoherence is presumably better approximated by the leading order effect of noise, beyond the static approximation, as discussed in the mentioned article. I suggest Authors to check.

In both cases (feedback off/on) the linewidths may more evidently point out the predicted decay laws. Spurious modes may be evidenced as well, as illustrated in fig 4 of G. Falci et al.

For a broader presentation of $1/f$ noise sources in superconducting qubits and decoherence effect evaluated using different approximation schemes I suggest Authors refer to E. Paladino, Y. Galperin, G. Falci, B. L. Altshuler Rev. Mod. Phys. 86, 361 (2014).

Additional questions, minor points and typos:

- Figure 2 a): Can Authors comment on the origin of the frequency mismatch between the excited state population without and with feedback (blue/orange lines)?
- I suggest defining explicitly the meaning of the quantities having a hat, like $\hat{\delta}$ and \hat{p}_1 .
- Line 322: "Fig 3 a)" has to be replaced with Fig. 3, please check.
- Line 691: Fig. 1b probably stays for Fig. 1D, please check.

Elisabetta Paladino

Reviewer #2 (Remarks to the Author):

In the work "Improving qubit coherence using closed-loop feedback", Vepsäläinen et. al. present a method for improving qubit coherence and single qubit gate fidelity using active feedback to monitor and correct the qubit control frequency. The technique involves interleaving experiments between Ramsey measurements on the scale of tens of microseconds using midcircuit

measurements and feedback on the qubit frequency. This enables the operation of a flux-tunable transmon qubit away from the sweet spot. The authors demonstrate an improvement in T_2 of 26%, from 6.2 μ s to 7.7 μ s.

This result does indicate a potential path towards overcoming some major challenges with using transmon qubits, which largely include balancing the tradeoff between frequency control for reducing collisions and flux noise. The ability to operate qubits far from the sweet spot relaxes the requirements on the frequency allocations of qubits in large arrays. These results also suggest the possibility of avoiding couplings to TLSs on the fly, which could be applicable more generally within superconducting qubit architectures.

While the experiments performed in this work are technically impressive, I don't find there is enough novelty or universal appeal to warrant publication in Nature Comms. The feedback technique itself is similar to other work cited in references 1-6 in the manuscript in both superconducting and other qubit architectures. In terms of calibrating drift in superconducting experiments there are also sophisticated techniques in the literature, such as the adaptive Floquet calibration in arXiv:2010.07965, although these do not use the closed-loop form of the present manuscript.

The improvements shown by closed-loop feedback in this paper are statistically significant, but they still yield modest coherence times that are still a fraction of the $2T_1$ limit. It seems likely that accessing a wider range of timescales in the Ramsey experiments would further improve the low noise suppression, but it's not clear that this would overcome the limitations flux noise puts on the coherence times of tunable qubits. The closed feedback alone does not fundamentally change the trade space for flux-tunable qubits.

The paper is generally well written and clear, and I find the results to be interesting. With superconducting qubits, noise can either be suppressed through new design and fabrication processes or through control techniques, and this is a nice entry in the control techniques category. The ability to interleave Ramsey sequences and frequency corrections on such a fast time scale is a new addition to this literature. However, I do not believe that these results meet the threshold for publishing in a broad interest journal like Nature Comms, and rather it is more suited to a specialized journal with a quantum focus.

Specific comments:

-Figure 1(b) is difficult to read. Perhaps this could be enlarged so that the text on the pulse labels is more legible, since this is a critical figure for understanding the experiment that is implemented.

-In the benchmarking plot in Figure 4(b), I would suggest extending these curves until they are fully decayed to ensure accurate fits and to see the final decay values. I don't think it's necessary to repeat this experiment for this paper, but it would be interesting to know, for example, that there is little or the same leakage in both feedback on/off cases. It also appears that the feedback off case starts from a lower value at zero Cliffords – is this actually indicated there is a higher excited state population to begin with when there is no feedback?

-The discussion section mentions that the open-loop feedback technique could be used to suppress noise sources other than $1/f$ noise, but it seems to be lacking any comment on how effectively it could suppress noise if you could access a wider range of the noise spectral density. What are the limitations on T_2 such that it is still less than $8\mu\text{s}$ instead of $40\text{-}80\mu\text{s}$ given by $2 \cdot T_1$? This seems to be an important question for understanding the significance of this result – does it really pave the way for large arrays of flux qubits with high fidelity gates?

Reviewer #3 (Remarks to the Author):

This work « Improving qubit coherence using closed-loop feedback » by Vepsäläinen et al. reports a careful investigation of a strategy for mitigating the unavoidable slow frequency drifts of flux-tunable transmon qubits. This issue is important because qubit frequency agility is needed in many processor architectures. The problem is often left aside in other works by reporting only echo coherence times in place of Ramsey coherence times. The experimental work is well done, the results are convincing and carefully analyzed. They show that the strategy implemented in the work does significantly improve transmon qubit coherence and gate performance. These results are of interest to the broad qubit community beyond transmon experts. Last but not least, the manuscript is well written and easy to follow. Here are a few remarks to be considered.

-The Figure 1 plays an important role for the description of the work, and displays an important part of the results in panel d). This rich panel should be larger for the sake of clarity. Panel a), at the opposite, could be reduced without information loss.

-The probing phase procedure is well described. The reader may wonder what happens if a sudden frequency jumps implies a change of the integer k in Eq (2). What happens ?

-The feedback efficiency falls down at 1 kHz, which explains why the gain in Ramsey coherence time is rather modest (26% quoted at line 245), even if a more spectacular x3 gain is obtained on averaged Ramsey sequences. The maximum gain one could expect from feedback would be to reach the dephasing noise cancellation of spin-echo. As shown in Fig 3, the present work falls short of approaching this limit because of limited feedback bandwidth. The feedback bandwidth issue is discussed on lines 330-335. Here, the reader wants to know if a larger bandwidth could be achieved, and what is the improvement margin for coherence times.

-Note that Figure (3) is quoted as Fig. 3a in line 322, but that there is no panel 3b or call to Fig. 3b in the text.

-The interest of the method is evaluated for single qubit gates using randomized benchmarking. Note that showing the position of 1 on the fidelity X axis of Fig 4d would help the reader to evaluate the gain achieved using feedback.

However, this does not tell if two-qubit gate fidelity would also be improved, and by how much.

Along the same lines, the text says, at line 443, that multi-qubit systems could benefit from feedback. Could the authors make a more precise statement on NISQ processors ? Could a new NISQ generation benefit from this form of closed-loop feedback ?

-The supplemental material is fine.

Referee #1

We thank the referee for the recommendation to publish, and for the detailed and helpful comments that have helped us improve the manuscript. Below we address the comments point by point.

In the following, **blue** indicates referee text, **black** indicates our response to the referee, and **red** indicates modifications made to the manuscript.

In the present manuscript Authors report the experimental realization of closed-loop feedback to stabilize low frequency fluctuations of the frequency of a superconducting flux tunable transmon qubit. This is a crucial issue since limitation in coherence times of present day superconducting qubits is a key limiting effect towards the realization of larger scale quantum processors. One of the main issues are charge, flux and Josephson energy noise with $1/f$ spectrum, a problem investigated in a number of articles (see below for relevant references on this topic). The key results of the manuscript are the demonstration of 26% improvement of the decoherence time away from the flux sweet point and the 99.9% fidelity of single-qubit randomized benchmarking.

Original results are derived using appropriate experimental methods and data are analyzed applying established frequency estimation methods to the closed-loop feedback protocol consisting of three phases that are continuously repeated. Obtained results, potentially allowing for increasing the operable frequency band of qubits, may have relevant implications for addressing the frequency crowding problem in large scale quantum processors. The manuscript is certainly suitable for publication in Nat. Comm., but it may still be improved considering the following issues.

The closed-feedback approach is essentially a method to recalibrate the inhomogeneous broadening effect of the qubit splitting due to repeated runs of an experiment taking place in the presence of low-frequency noise. The quasi-static approximation captures this effect and results in short-times decoherence of Ramsey fringes with a characteristic non-exponential decay, depending on the operating point, and obtained originally in G. Falci et al Phys Rev Lett 94, 167002 (2005) and shortly later in Ref.[27] of the manuscript. In G. Falci et al, the improvement by recalibration was demonstrated and a systematic inclusion of the effect of higher-frequency noise components, active with feedback-on, was performed, leading to analytical predictions.

We thank the reviewer for the relevant reference, which provides a theoretical framework for studying the effects of recalibration – a concept which is closely linked to our feedback implementation. We have added a citation in the manuscript line 113.

To analyze dephasing, we have used the approximate result which does not include the noise at qubit frequency. This approximation is widely used for transmon qubits because flux noise mainly couples through σ_z operator, resulting in $\theta \approx 0$. However, we find the analysis in the article very interesting and relevant for the analysis of some other superconducting qubit modalities, such as the flux qubit.

I expect that the present work may be strengthened by the following analysis of experimental data.

- Case feedback off: the accuracy level of the present experiment may provide a benchmark of the power law to exponential quadratic decay predicted in G. Falci et. al, at- and away-from the flux optimal point. This analysis may also point out whether noise in the Josephson energy plays a role, as pointed out in Ref. [14] of the manuscript and which I would have expected to observe.

Away from the sweetspot, we noticed that quadratic exponential decay does not describe the decay envelope very well, but we need to use Eq. (5) with the full flux-noise spectral density instead. Near the sweet spot the flux-noise contribution vanishes, revealing the exponential decay due to T1.

As suggested, we checked whether noise in the Josephson energy could play a role in our setup. According to Ref. [14] dephasing time due to fluctuations in the critical current scale as f_q^{-1} , which is opposite from what we observe when we tune the qubit frequency (See black line in Fig. 3). From that we conclude that the dephasing away from the sweet spot is dominated by flux noise. At the sweet spot the first order flux noise is suppressed, and the remaining contributing noise sources such as photon shot noise from the readout resonator, Josephson current noise, and second order flux noise dictate the dephasing.

- Case feedback on: One of the key results of the present manuscript is the improvement of decoherence times away from the qubit flux-noise insensitive point obtained by closed-feedback. This effect is analysed based on eq. (5) which holds under pure dephasing and it is used to fit data in Fig. 3 with a reasonable but not excellent agreement. Under the considered conditions the residual decoherence is presumably better approximated by the leading order effect of noise, beyond the static approximation, as discussed in the mentioned article. I suggest Authors to check.

As the reviewer writes, in Fig. 3 we extracted the coherence times from a fit to Eq. (5), but we also included an exponential decay component corresponding to T1 of the qubit. We note that Eq. (5) goes beyond the quasi-static approximation by including the full spectral density of the noise. Near the sweet spot Eq. (5) is no longer strictly valid due to contribution of higher order noise spectra – a signature of violation of the Gaussian approximation. Nevertheless, we can extract the noise amplitude k from the linear fit to the data as a function of the qubit flux sensitivity.

As the reviewer pointed out, the measured dephasing times do not fall perfectly on a line, which we attribute to low frequency noise induced by parasitic two-level fluctuators. Note that the dephasing times measured using the spin-echo sequence fit very well to the linear noise model, indicating the presence of low-frequency noise from other sources than flux noise at certain qubit frequencies. At those frequencies, we also observe either reduced or fluctuating gate fidelity.

In both cases (feedback off/on) the linewidths may more evidently point out the predicted decay laws. Spurious modes may be evidenced as well, as illustrated in fig 4 of G. Falci et al.

For a broader presentation of $1/f$ noise sources in superconducting qubits and decoherence effect evaluated using different approximation schemes I suggest Authors refer to E. Paladino, Y. Galperin, G. Falci, B. L. Altshuler Rev. Mod. Phys. 86, 361 (2014).

We thank the reviewer for the suggestion of analyzing the decay laws in Fourier space, and for providing the reference. We have added a citation to the review article in the introduction. Following the suggestion, we have analyzed the linewidths of the Ramsey experiments with and without feedback (see the attached figure below). At the sweet spot the feedback is not effective due to qubit being insensitive to flux-based feedback control. As the qubit is tuned away from the sweet spot, we noticed that at some frequencies there is beating in the Ramsey oscillations, which is clearly visible in the Fourier transform (lower panels). This aligns well with the bi-stable fluctuator model from Phys Rev Lett 94, 167002 (2005). Note that feedback is able to suppress the splitting in the spectrum, demonstrating that feedback is effective at suppressing low-frequency noise from spectrally active parasitic modes.

Questions, minor points and typos:

- Figure 2 a): Can Authors comment on the origin of the frequency mismatch between the excited state population without and with feedback (blue/orange lines)?

The frequency mismatch is a result of the feedback locking the qubit frequency to exactly to the target value. Without feedback, the detuning from the drive tone slightly drifts over time, which results in different Ramsey frequencies with and without the feedback.

- I suggest defining explicitly the meaning of the quantities having a hat, like $\hat{\Delta}$ and \hat{p}_1 .

Thank you for the suggestion. To clarify the definitions, we have added explicit reference to Eq. (2) on line 123 where the estimator variables have been defined.

- Line 322: “Fig 3 a)” has to be replaced with Fig. 3, please check.

Thank you for spotting this!

- Line 691: Fig. 1b probably stays for Fig. 1D, please check.

That’s right. Thanks you!

Elisabetta Paladino

Reviewer #2 (Remarks to the Author):

In the following, **blue** indicates referee text, **black** indicates our response to the referee, and **red** indicates modifications made to the manuscript.

We thank the referee and taking the time to read our work in detail and for raising several important points. Below, we address the referee comments point by point.

In the work “Improving qubit coherence using closed-loop feedback”, Vepsäläinen et. al. present a method for improving qubit coherence and single qubit gate fidelity using active feedback to monitor and correct the qubit control frequency. The technique involves interleaving experiments between Ramsey measurements on the scale of tens of microseconds using midcircuit measurements and feedback on the qubit frequency. This enables the operation of a flux-tunable transmon qubit away from the sweet spot. The authors demonstrate an improvement in T_2 of 26%, from 6.2 μ s to 7.7 μ s.

This result does indicate a potential path towards overcoming some major challenges with using transmon qubits, which largely include balancing the tradeoff between frequency control for reducing collisions and flux noise. The ability to operate qubits far from the sweet spot relaxes the requirements on the frequency allocations of qubits in large arrays. These results also suggest the possibility of avoiding couplings to TLSs on the fly, which could be applicable more generally within superconducting qubit architectures.

While the experiments performed in this work are technically impressive, I don’t find there is enough novelty or universal appeal to warrant publication in Nature Comms. The feedback technique itself is similar to other work cited in references 1-6 in the manuscript in both superconducting and other qubit architectures. In terms of calibrating drift in superconducting experiments there are also sophisticated techniques in the literature, such as the adaptive Floquet calibration in arXiv:2010.07965, although these do not use the closed-loop form of the present manuscript.

We would like to highlight the advantages of our work compared to the ones in the references. Our protocol is directly applicable for improving the coherence time of arbitrary gates, which has not been realized in superconducting qubit architecture by any of the feedback papers. Ref. [6] uses a spin-qubit to implement an algorithm that is

quite similar to ours, and they successfully demonstrate an improved gate fidelity. However, their noise spectrum is mostly $1/f^2$ which does not require high-bandwidth for the feedback, and they do not reach state-of-the-art gate fidelity compared to other implementations in their platform.

We agree that the Floquet calibrating method developed by the Google team is very powerful. Our work significantly differs from theirs in that all the calculations required for the feedback are done in FPGA hardware, which enables 70 microsecond calibration time as opposed to about 1 minute quoted in arXiv:2010.07965 (~6 orders of magnitude difference in time scales). The difference in time scales is crucial for our results, as it determines the bandwidth at which the noise is suppressed by the feedback and thereby dictates how much the dephasing time is improved. The high bandwidth is especially important for suppressing noise with a $1/f^\alpha$ spectrum, where $\alpha \sim 1$ – a typical value for flux noise – because in practice a significant fraction of the noise power comes from frequencies above $(1 \text{ min})^{-1}$.

The improvements shown by closed-loop feedback in this paper are statistically significant, but they still yield modest coherence times that are still a fraction of the $2T_1$ limit. It seems likely that accessing a wider range of timescales in the Ramsey experiments would further improve the low noise suppression, but it's not clear that this would overcome the limitations flux noise puts on the coherence times of tunable qubits. The closed feedback alone does not fundamentally change the trade space for flux-tunable qubits.

The maximal performance ultimately depends on the functional form of the noise power spectral density function. $1/f^2$ can be suppressed almost completely, whereas $1/f$ noise is much more complicated. In a reply to a following comment, we show that under ideal conditions it would be possible to increase the coherence time of the qubit by a factor of 3.7. Reaching that level of noise suppression in experiments would require some adjustments to the feedback algorithm as well as improved readout speed, but we see no fundamental limitations to reaching that goal. Improving coherence time by a factor of 3.7 would result in more than an order of magnitude improvement in gate fidelities if dephasing is the limiting factor.

In addition to improving the coherence times, our implementation is also able to significantly mitigate slow frequency drifts and frequency fluctuation caused by parasitic interaction with two-level fluctuators. In our experiments, we found this feature to have a huge practical significance, and it allowed us to operate the qubit at operation points that without feedback were not feasible. For these reasons, we believe that our feedback implementation could play a crucial role in helping to mitigate frequency crowding issues in large qubit grids – a major issue in contemporary devices.

The paper is generally well written and clear, and I find the results to be interesting. With superconducting qubits, noise can either be suppressed through new design and fabrication processes or through control techniques, and this is a nice entry in the control techniques category. The ability to interleave Ramsey sequences and frequency corrections on such a fast time scale is a new addition to this literature. However, I do not believe that these results meet the threshold for publishing in a broad interest journal like Nature Comms, and rather it is more suited to a specialized

journal with a quantum focus.

We thank the reviewer for the useful comments and the positive evaluation. We hope the reviewer finds our replies satisfying.

Specific comments:

-Figure 1(b) is difficult to read. Perhaps this could be enlarged so that the text on the pulse labels is more legible, since this is a critical figure for understanding the experiment that is implemented.

We thank the reviewer for the comment. We have increased the font sizes in Fig. 1b and 1d.

-In the benchmarking plot in Figure 4(b), I would suggest extending these curves until they are fully decayed to ensure accurate fits and to see the final decay values. I don't think it's necessary to repeat this experiment for this paper, but it would be interesting to know, for example, that there is little or the same leakage in both feedback on/off cases. It also appears that the feedback off case starts from a lower value at zero Cliffords – is this actually indicated there is a higher excited state population to begin with when there is no feedback?

We agree that it would have been optimal to use slightly longer RB sequences to fully see the final decay value.

We checked by comparing to other data sets that the fluctuation in the initialization fidelity does not depend if we use feedback or not. The initial population fluctuates from experiment to experiment by approximately ± 0.02 . This does not necessarily imply that the initial excited state population of the qubit would fluctuate. In the virtual reset protocol we use (introduced on line 127 of the manuscript) we do not physically initialize the qubit in the ground state between the measurements, but use the measured state after the previous experiment to deduce the initial state. As a result, the initialization fidelity is affected by the readout fidelity, which is the most likely cause for the fluctuations.

-The discussion section mentions that the open-loop feedback technique could be used to suppress noise sources other than $1/f$ noise, but it seems to be lacking any comment on how effectively it could suppress noise if you could access a wider range of the noise spectral density. What are the limitations on T_2 such that it is still less than $8\mu\text{s}$ instead of $40\text{-}80\mu\text{s}$ given by $2 \cdot T_1$? This seems to be an important question for understanding the significance of this result – does it really pave the way for large arrays of flux qubits with high fidelity gates?

This is indeed a very important question, and we try to answer to it in detail. We have added a section “**Analysis on fundamental limitations of the feedback implementation**” in the Supplementary material where we analyze the reachable optimal performance of the feedback algorithm. For convenience, we briefly review the results here.

The noise suppression is most efficient for noise spectra which have a lot of weight below the bandwidth of the feedback. For example, $1/f^2$ noise can be suppressed very efficiently whereas only a minor fraction of white noise can be removed. Flux noise is mostly $1/f$ noise, and falls somewhere in between. In the Supplementary material we now show that under ideal conditions, the coherence time of the qubit could be increased by a factor of 3.7 (270 % improvement) using our implementation if the noise spectral density from Fig. 1d and cutoff frequency from Fig. 2a is assumed. While reaching that level of improvement in the experiment requires improvements to our hardware implementation and readout speed, we don't see any fundamental limitations preventing us from reaching that in the future experiments. The figure below shows the simulation of the feedback performance as a function of the Ramsey delay τ with number of Ramsey experiments per frequency estimate fixed $N = 1$ (left) or $N = 20$ (right). For additional details, see added section in the Supplementary material.

the simulation for $1/f^2$ noise spectral density to demonstrate that feedback is very effective in mitigating certain common noise spectra. In this case, the coherence time is improved by more than a factor of 1000 (figure below).

We believe that these simulations demonstrate that the feedback protocol is a powerful tool with potential for significantly improving coherence times of qubits and their gate fidelities. For example, for qubits which gate fidelity is limited by dephasing, factor of 3.7 improvement in coherence time would result in a reduction in gate errors by a factor of $3.7^2 \approx 14$.

Reviewer #3 (Remarks to the Author):

We thank the referee for detailed comments, which have helped us to improve the manuscript, and his/her positive evaluation. Below we respond to all of the referee comments in detail.

In the following, **blue** indicates referee text, **black** indicates our response to the referee, and **red** indicates modifications made to the manuscript.

This work « Improving qubit coherence using closed-loop feedback » by Vepsäläinen et al. reports a careful investigation of a strategy for mitigating the unavoidable slow frequency drifts of flux-tunable transmon qubits. This issue is important because qubit frequency agility is needed in many processor architectures. The problem is often left aside in other works by reporting only echo coherence times in place of Ramsey coherence times. The experimental work is well done, the results are convincing and carefully analyzed. They show that the strategy implemented in the work does significantly improve transmon qubit coherence and gate performance. These results are of interest to the broad qubit community beyond transmon experts. Last but not least, the manuscript is well written and easy to follow. Here are a few remarks to be considered.

-The Figure 1 plays an important role for the description of the work, and displays an important part of the results in panel d). This rich panel should be larger for the sake of clarity. Panel a), at the opposite, could be reduced without information loss.

We agree on this with the reviewer. We have enlarged the right panels and shrunk the left panels to give d) a little bit more space.

-The probing phase procedure is well described. The reader may wonder what happens if a sudden frequency jumps implies a change of the integer k in Eq (2). What happens ?

This is an interesting question, and an important factor in the optimization/application of the algorithm. If the integer k changes due to a sudden frequency jump the feedback locks in a wrong minimum, resulting in a constant offset in the qubit frequency. Figure below shows a repeated Ramsey experiment (this is a different dataset from Fig. 2c), where τ was chosen to be too high, and feedback occasionally locked at wrong frequency due to a rapid change in the qubit frequency.

In the experiment, we avoid accidentally locking to a wrong frequency by using Ramsey delay τ that provides high enough noise amplitude bandwidth. In addition to controlling the amplitude bandwidth by reducing τ , the frequency bandwidth also plays an important role in preventing incorrect frequency lock. Since we adjust the qubit frequency at every step of the feedback, the integrated noise power from step to step depends on the feedback rate.

We are working on an FPGA implementation of a more advanced version of the feedback algorithm, which would allow us to use two values of τ to be able to increase sensitivity of the algorithm while maintaining the robustness to sudden frequency jumps.

-The feedback efficiency falls down at 1 kHz, which explains why the gain in Ramsey coherence time is rather modest (26% quoted at line 245), even if a more spectacular x3 gain is obtained on averaged Ramsey sequences. The maximum gain one could expect from feedback would be to reach the dephasing noise cancellation of spin-echo. As shown in Fig 3, the present work falls short of approaching this limit because of limited feedback bandwidth. The feedback bandwidth issue is discussed on lines 330-335. Here, the reader wants to know if a larger bandwidth could be achieved, and what is the improvement margin for coherence times.

To answer to this question, we have added a section “**Analysis on fundamental limitations of the feedback implementation**” in the Supplementary material and refer to that in the main text. For convenience, we briefly review the results here.

In the figure below we have simulated the expected improvement in the coherence time as a function of the Ramsey delay τ of the frequency probing sequence. In addition to the bandwidth, we include the expected statistical sampling noise, which

increases the noise seen by the qubit. For $\tau = 15 \mu\text{s}$, the coherence time is improved by a factor of 3.7 due to the feedback (solid orange line, left panel). For lower values of τ , the statistical sampling noise reduces the efficiency, and for higher values the bandwidth of the feedback is reduced, resulting in sub-optimal performance. In the simulation we use the same cutoff parameters and noise spectral density as in Fig. 2a. The predicted optimal performance is very close to the measured echo dephasing time $\approx 25 \mu\text{s}$ at that operation point (see Fig. 3 at 4.690 GHz).

the bandwidth was limited because we had to use values of τ that were significantly lower than the optimal value due to the occasional rapid jumps in the frequency. To counteract the statistical sampling noise caused at low τ , we used the outcomes of $N=20$ repetitions of the measurement for a single estimate of the qubit frequency (see panel on the right for a simulation for $N=20$). As mentioned in the answer to the previous comment, we are working on an implementation that allows us to use multiple values of τ in order to increase both the amplitude and frequency bandwidth of the feedback algorithm. Another important objective is to reduce the readout duration and other delays in order to reduce the overall overhead. We are working on these issues on a separate project. We believe that these improvements will allow us to reach higher bandwidth in the future.

-Note that Figure (3) is quoted as Fig. 3a in line 322, but that there is no panel 3b or call to Fig. 3b in the text.

Thank you for pointing this out. We have corrected this.

-The interest of the method is evaluated for single qubit gates using randomized benchmarking. Note that showing the position of 1 on the fidelity X axis of Fig 4d would help the reader to evaluate the gain achieved using feedback.

We adjusted the axes in Fig. 4d to include 1.

However, this does not tell if two-qubit gate fidelity would also be improved, and by how much.

We agree that it would be interesting to see how much two-qubit gates can be improved using the feedback protocol. We are currently working on an experiment where we apply a slightly modified feedback protocol to improve a CPHASE gate fidelity realized with a tunable coupler.

Along the same lines, the text says, at line 443, that multi-qubit systems could benefit from feedback. Could the authors make a more precise statement on NISQ processors ? Could a new NISQ generation benefit from this form of closed-loop feedback ?

Our feedback implementation only relies on single qubit operations, which means that it can be easily operated in parallel on multi-qubit systems. As the number of qubits grow, the probability of at least few of them interacting with parasitic two-level fluctuators or suffering from drift increases dramatically. In those cases, we believe the proposed closed-loop feedback algorithm could prove particularly useful.

In the end of discussion, we have added the following:

“As quantum processors grow in size, the proposed feedback algorithm could increase their reliability by continuously calibrating for drifts in the individual qubits.”

-The supplemental material is fine.

REVIEWERS' COMMENTS

Reviewer #1 (Remarks to the Author):

Authors have satisfactorily addressed in their reply the issues raised in my report and added relevant references to the revised manuscript. I recommend for publication in Nature Communications.

As an optional suggestion, I invite authors to include in the Supplementary Material the considerations on the effectiveness of the considered feedback protocol for suppressing also low-frequency noise from spectrally active parasitic modes, as evinced from the Fourier analysis of the Ramsey experiment with and without feedback.

Finally, please check the following possible typos:

- Caption of Fig. 3, line 288: "times" to be replaced with "rates"?
- Line 312: Fig.3a to be replaced with Fig. 3.

Elisabetta Paladino

Reviewer #2 (Remarks to the Author):

I appreciate the authors' thoughtful response and the additions they have made to the paper. As stated before, the control hardware and techniques demonstrated in the paper are impressive. The changes strengthen their results. I still find the dephasing time improvements to be modest and believe further work will be needed to demonstrate this is a viable path towards avoiding frequency collisions. However I am convinced by the arguments that this is a significant step. Upon rereading, I am also not sure $2 \cdot T_1$ was a fair reference point given that these are T_2 times measured from Ramsey experiments and could be reported as T_2^* . This is not directly comparable to other reported values of T_2 that are typically measured with a Hahn echo. I support publication.

Reviewer #3 (Remarks to the Author):

The authors have understood the rather converging critics made by the referees. They have corrected small details and added a new section in the supplementary material. Although the coherence time improvement achieved in this work is rather modest, the method implemented for controlling qubit frequency is a very nice piece of work of interest for a broad audience. The revised version deserves publication in Nature Communications.

We thank all the reviewers for their detailed and useful feedback as well as the recommendation to publish in Nature Communications. Following the suggestions by reviewer 1, we have added a short section to the supplementary material discussing the impact of feedback on mitigating the beating due TLSs. We also corrected the typo's graciously pointed out by reviewer 1.

REVIEWERS' COMMENTS

Reviewer #1 (Remarks to the Author):

Authors have satisfactorily addressed in their reply the issues raised in my report and added relevant references to the revised manuscript. I recommend for publication in Nature Communications.

As an optional suggestion, I invite authors to include in the Supplementary Material the considerations on the effectiveness of the considered feedback protocol for suppressing also low-frequency noise from spectrally active parasitic modes, as evinced from the Fourier analysis of the Ramsey experiment with and without feedback.

Finally, please check the following possible typos:

- Caption of Fig. 3, line 288: "times" to be replaced with "rates"?
- Line 312: Fig.3a to be replaced with Fig. 3.

Elisabetta Paladino

Reviewer #2 (Remarks to the Author):

I appreciate the authors' thoughtful response and the additions they have made to the paper. As stated before, the control hardware and techniques demonstrated in the paper are impressive. The changes strengthen their results. I still find the dephasing time improvements to be modest and believe further work will be needed to demonstrate this is a viable path towards avoiding frequency collisions. However I am convinced by the arguments that this is a significant step. Upon rereading, I am also not sure $2 \cdot T_1$ was a fair reference point given that these are T_2 times measured from Ramsey experiments and could be reported as T_2^* . This is not directly comparable to other reported values of T_2 that are typically measured with a Hahn echo. I support publication.

Reviewer #3 (Remarks to the Author):

The authors have understood the rather converging critics made by the referees. They have corrected small details and added a new section in the supplementary material. Although the coherence time improvement achieved in this work is rather modest, the method implemented for controlling qubit frequency is a very nice piece of work of

interest for a broad audience. The revised version deserves publication in Nature Communications.